# Indole Alkaloids and Chromones from the Stem Bark of *Cassia alata* and Their Antiviral Activities

**DOI:** 10.3390/molecules27103129

**Published:** 2022-05-13

**Authors:** Pei-Song Yang, Jia-Meng Dai, Xue-Jiao Gu, Wen Xiong, De-Quan Huang, Shi-Yu Qiu, Jun-Na Zheng, Yong Li, Feng-Xian Yang, Min Zhou

**Affiliations:** 1Key Laboratory of Chemistry in Ethnic Medicinal Resources, State Ethnic Affairs Commission & Ministry of Education, Yunnan Minzu University, Kunming 650031, China; sgyangpeisong@163.com (P.-S.Y.); gxj15025127545@163.com (X.-J.G.); 18397968317@163.com (D.-Q.H.); qshiyu2022@163.com (S.-Y.Q.); z979117276@163.com (J.-N.Z.); 2Yunnan Key Laboratory of Tobacco Chemistry, China Tobacco Yunnan Industrial Co., Ltd., Kunming 650231, China; daijiameng99@163.com (J.-M.D.); ldxwen@126.com (W.X.); liyong_yx@163.com (Y.L.)

**Keywords:** *Cassia alata*, indole alkaloids, chromones, anti-TMV activities, anti-rotavirus activities

## Abstract

The *Cassia* (Leguminosae) genus has attracted a lot of attention as a prolific source of alkaloids and chromones with diverse structures and biological properties. The aim of this study is to screen the antiviral compounds from *Cassia alata*. The extract of the stem bark of this plant was separated using silica gel, MCI, ODS C18, and Sephadex LH-20 column chromatography, as well as semi-preparative HPLC. As a result, three new indole alkaloids, alataindoleins A–C (1–3); one new chromone, alatachromone A (4); and a new dimeric chromone-indole alkaloid, alataindolein D (5) were isolated. Their structures were determined by means of HRESIMS and extensive 1D and 2D NMR spectroscopic studies. Interestingly, alataindolein D (5) represents a new type of dimeric alkaloid with an unusual N-2−C-16’ linkage, which is biogenetically derived from a chromone and an indole alkaloid via an intermolecular nucleophilic substitution reaction. Compounds 1–5 were tested for their anti-tobacco mosaic virus (TMV) and anti-rotavirus activities, and the results showed that compounds 2–4 showed high anti-TMV activities with inhibition rates of 44.4%, 66.5%, and 52.3%, respectively. These rates were higher than those of the positive control (with inhibition rate of 32.8%). Compounds 1 and 5 also showed potential anti-TMV activities with inhibition rates of 26.5% and 31.8%, respectively. In addition, compounds 1–5 exhibited potential anti-rotavirus activities with therapeutic index (TI) values in the range of 9.75~15.3. The successful isolation and structure identification of the above new compounds provided materials for the screening of antivirus drugs, and contributed to the development and utilization of *C. alata*.

## 1. Introduction

*Cassia* Linn, a cosmopolitan genus of the legume family (Fabaceae) includes 600 species, some of which (*C. siamea*, *C. occidentalis*, *C. obtusifolia*, and *C*. *alata*, etc.) are used in the folk medicine of the Chinese Dai tribe, notably for the treatment of rheumatoid arthritis, skin diseases, diabetes, malaria, skin trauma and constipation [1,2,3]. Among them, *C. alata* locally known as “Ya-La-Meng-Long”, is commonly used as an effective treatment for hypertension, some gastrointestinal diseases, and skin diseases in clinics and drug stores of the subtropical region of Yunnan Province, China [3,4,5].

Previous phytochemical investigations on this genus revealed several novel bioactive alkaloids, such as cassiarin A [6], a promising antiplasmodial tricyclic alkaloid; fistulain A, possessing a new type of dimeric chromone alkaloid consisting of a chromone and cassiarin A [7]; and cassiarin F, a novel hybrid alkaloid, which was biogenetically derived from cassiarin A and a biphenyl unit with an acetonyl moiety [8]. In our previous works, some bioactive metabolites, such as chromones [9], hetero-dimeric polyphenols [10], flavones [11], alkaloids [12], and anthraquinones [13] were also isolated from *C. alata*.

In our ongoing research on biologically active metabolites from the genus *Cassia*, we are reinvestigating the chemical constituents of the stem bark of *C. alata* collected in Yingjiang County, Dehong Prefecture, which led to the isolation and identification of three new indole alkaloids, alataindoleins A–C (1–3); one new chromone, alatachromone A (4); and a new dimeric chromone-indole alkaloid, alataindolein D (5). Among them, alataindolein D, a novel-type dimeric alkaloid, was biosynthesized from an indole alkaloid and a chromone with an unusual coupling pattern (N-2−C-16’ linkage). In addition, compounds 2–4 showed high anti-TMV activities, and compounds 1–5 also exhibited potential anti-rotavirus activities. Herein, we report the isolation and structural elucidation of the above new compounds, as well as their antivirus properties.

## 2. Results and Discussion

A 70% aq. acetone extract prepared from the stem barks of *C. alata* was partitioned between EtOAc and 3% tartaric acid. The EtOAc-soluble material was repeatedly subjected to column chromatography and preparative HPLC to yield one new chromone, alatachromone A (4), together with three known chromones (7–9). Then, the above aqueous layer was adjusted to pH 10 with saturated NaOH aq. and extracted with EtOAc again. The EtOAc-soluble alkaloidal materials were repeatedly subjected to column chromatography on silica gel and preparative HPLC to yield four new alkaloids, alataindoleins A–D (1–3, 5), together with one known alkaloid (6). The structures of the isolated compounds are shown in Figure 1, their ^1^H and ^13^C NMR data are listed in Table 1, Table 2 and Table 3, and the Figures for NMR see Appendix A. By a comparison with other studies, the known compounds were identified as 5-(hydroxymethyl)-2-methyl- 6-prenylisoindolin-1-one (6) [14], 8-(3-hydroxy- propyl)-2,2,6-trimethylchroman-4-one (7) [15], 5-methoxy-2,2-dimethyl-7-(2-oxopropyl)-chroman-4-one (8) [16], and 5-methoxy- 2,2-dimethyl-8-(2-oxopropyl)-2,3-dihydro- chromen-4-one (9) [16]. To the best of our knowledge, compound 5 represents a novel type of dimeric alkaloid, which might be biosynthetically formed via an intermolecular nucleophilic substitution reaction between an indole alkaloid and a chromone.

### 2.1. Structure Elucidation

Compound 1 was isolated as yellow oil. The molecular formula of 1 was determined to be C_11_H_13_NO_2_ by the molecular ion peak at *m/z* 214.0851 [M+Na]^+^ in its HRESIMS, suggesting 6 degrees of unsaturation. The UV spectrum showed absorption maxima at 210, 255 and 292 nm, and the IR spectrum showed absorption bands at 3084, 2946, 1658, 1614, 1568, 1459 cm^−1^, indicating the presence of carbonyl and aromatic rings. The ^1^H, ^13^C, and HSQC NMR data (Table 1) of 1 show resonances due to a 1,2,3,5-tetrasubstituted benzene ring (C-4~C-9, H-5, and H-7), an acyl carbon (C-1), a methylene carbon (C-3 and H_2_-3), one methoxy group (*δ*_C_ 56.2 q, *δ*_H_ 3.78 s), and two methyl groups (C-10, C-11, H_3_-10, and H_3_-11). Based on its molecular formula, the acyl and methylene carbons should be connected by a nitrogen atom to form an isoindolin-1-one nucleus [17,18] to support the 6 degrees of unsaturation. This deduction was also supported by the HMBC correlations (Figure 2) from H_2_-3 to C-1, C-4, C-8, and C-9, and from H-7 to C-1, C-8, and C-9. Since the nucleus of compound was determined, the additional carbons (two methyl and one methoxy group) accounted for the remaining substituents. The location of the methoxy group was assigned to the C-4 position based on the HMBC correlations from methoxy protons (*δ*_H_ 3.78) to C-4. Two methyl groups located at C-6 and N-2 were supported by the HMBC correlations from H_3_-11 to C-5, C-6, and C-7; from H-5 and H-7 to C-11; from H_3_-10 to C-1, C-3; and from H_2_-3 to C-10, respectively. Thus, the structure of 1 was established as 4-methoxy-2,6-dimethyl- isoindolin-1-one, and it was given a trivial name of alataindolein A. 

Compound 2 was also obtained as yellow gum. A molecular formula C_11_H_13_NO_3_ was assigned from HRESIMS (*m/z*: 230.0790 [M+Na]^+^, calcd 230.0793). The ^1^H and ^13^C NMR data of 2 (Table 1) were highly similar to those of 1. The obvious differences resulted from the replacement of the methyl group in 1 by the hydroxylmethyl group in 2. The location of the hydroxylmethyl group at C-6 was also supported by the HMBC correlations of the hydroxylmethyl proton (H_2_-11) with C-5, C-6, and C-7, of H-5, and H-7 with C-11. Accordingly, the structure of 6-(hydroxymethyl)-4-methoxy-2- methylisoindolin-1-one (2) was established, and its trivial name became alataindolein B.

Regarding 6-(3-Hydroxypropyl)-4-methoxy-2-methylisoindolin-1-one (3), a yellow gum, its ^1^H and ^13^C NMR spectra data were also similar to those of 1. Chemical shift differences resulted from the disappearance of a methyl group and the appearance of 3-hydroxypropyl group signals (-CH_2_CH_2_CH_2_-OH, C-11~C-13, H_2_-11~H_2_-13) [17,19] in compound 3. This indicated that the methyl group at C-6 in 1 was converted into a 3-hydroxypropyl group in compound 3. The HMBC correlations of H_2_-11 with C-5, C-6, and C-7, of H_2_-12 with C-6 also indicated this structure change. Thus, the structure of 3 was established as shown, and it was given a trivial name of alataindolein C.

Compound 4 was obtained as pale yellow gum after being purified by preparative HPLC. The compound gave a parent ion by HR-MS at *m/z* 261.1120 [M-H]^-^ (calculated for C_15_H_17_O_4_, 261.1127), corresponding to a molecular formula C_15_H_18_O_4_, and requiring seven degrees of unsaturation. The ^1^H and ^13^C NMR spectrum of compound 4, along with an analysis of the DEPT spectra (Table 2), displayed 15 carbon signals and 18 proton signals, respectively. These signals corresponded to a 1,2,3,4-tetrasubstituted benzene ring (C-5~C-10, H-5, and H-6, including an oxidized carbon), one 2-oxopropyl group (-CH_2_CO-CH_3_, C-11~C-13, H_2_-11, H_3_-13) [16,20], one hydroxymethyl group (C-16, H_2_-16), one *gem*-dimethyl carbon (C-14,15 and H_6_-14,15), one carbonyl carbon (C-4), one methylene carbon (C-3 and H_2_-3), and one quaternary carbon (C-2). Strong absorption bands accounting for the hydroxyl group (3409 cm^−1^), carbonyl group (1718 and 1672 cm^−1^), and aromatic groups (1618, 1550, and 1439 cm^−1^) could also be observed in its IR spectrum. The UV spectrum of 4 showed absorption maxima at 354, 257, and 212 nm, which confirmed the existence of aromatic functions. These aromatic functions were also present on the benzene ring in the molecule to support the seven degrees of unsaturation, and the typical signals for one oxidized carbon on the benzene ring (C-9), carbonyl (C-4), *gem*-dimethyl (C-14,15), methylene (C-3), and quaternary carbon (C-2) should be formed a *gem*-dimethyl chromone nucleus. This deduction could also be supported by the HMBC correlations (Figure 2) from H-5 to C-4, C-9, and C-10 from H-3 to C-10. Since the nucleus of the compound was determined, the additional carbons (2-oxopropyl and hydroxymethyl group) accounted for the remaining substituents. The HMBC correlations (Figure 2) of H_2_-11 with C-7, C-8 and C-9 indicated that the 2-oxopropyl group should be located at C-8 on the chromone ring. The hydroxymethyl group located at C-7 was supported by the HMBC correlations from H_2_-16 to C-6, C-7, and C-8, and from H-6 to C-16. Thus, the structure of 4 was established as shown. It was given the systematic name of 7-(hydroxymethyl)- 2,2-dimethyl-8-(2-oxopropyl)chroman-4-one, and the trivial name of alatachromone A.

Compound 5 was obtained as yellow gum. Its molecular formula C_25_H_27_NO_5_ was established from HRESIMS (*m/z* at 444.1782 [M+Na]^+^), possessing an index of hydrogen deficiency of 13. The IR spectrum showed absorption bands due to carbonyl (1726, 1675, 1658 cm^−1^) and aromatic groups (1614, 1574, 1538, and 1462 cm^−1^). The UV spectrum of compound 5 showed absorption maxima at 360, 268, and 216 nm, which confirmed the existence of the aromatic functions. The ^1^H NMR spectrum displayed signals of four methyl groups (H_3_-10, H_3_-13’, and H_6_-14’,15’), four methylene groups (H_2_-3, H_2_-3’, H_2_-11’, and H_2_-16’), four uncoupled aromatic protons (H-5, H-7, H-5’, and H-6’), and one methoxy proton (*δ*_H_ 3.78 s). The 25 carbon resonances observed in the ^13^C NMR and DEPT spectra (Table 3) were assignable to four methyls, four methylenes, four aromatic methines, a methoxy (*δ*_C_ 56.2 q), and twelve quaternary carbons (including three carbonyls and eight aromatic carbons). Among them, three carbonyls and twelve olefinic carbons account for 8 degrees of unsaturation, suggesting that compound 5 is a highly aromatized C-25 nitrogen-containing structure with a tetracyclic ring system. The ^1^H and ^13^C NMR spectra data of compound 5 were highly similar to those of 1 in C-1~C-10 (part a), and highly similar to those of compound 4 in C-2’~C-15’ (part b). The obvious chemical shift differences resulted from the disappearance of a *N*-methyl group resonance (C-10, H_3_-10) in 1 and hydroxymethyl group (C-16, H_2_-16) group in compound 5, and appearance of a *N*-methylene (C-16’, H_2_-16’). The above information indicated that compound 5 should be a heterodimer comprising a benzoisoindolin-1-one moiety (part a) and a *gem*-dimethyl chromone moiety (part b), and two moieties were connected by a *N*-atom and C-16’. The connection from C-16’ to the *N*-atom was supported by the HMBC correlations (Figure 2), which formed H_2_-16’ to C-3, C-1, C-6’, C-7’, and C-8’; from H-6’ to C-16’; and from H-3 to C-16’. The existence of *gem*-dimethyl chromone, benzoisoindolin-1-one and the substituent positions, can also be confirmed by a further analysis of its HMBC correlations. Accordingly, the structural assignment of 5 is depicted in Figure 1 and given a trivial name of alataindolein D. To the best of our knowledge, this compound has a new carbon skeleton with a N-2−C-16’ linkage, which formed between an indole alkaloid and a chromone via an intermolecular nucleophilic substitution reaction (Figure 3).

### 2.2. Anti-TMV Activities

Since certain alkaloids and chromones from the *Cassia* genus exhibit potential anti-TMV activities [7,10,12,15,16,17,18], the anti-TMV activities for compounds 1–5 were tested using the half-leaf method [21,22] at a concentration of 20 μM. Ningnanmycin (a commercial product for plant disease in China, with inhibition rate of 32.8%) was used as a positive control. The results (Table 4 and Appendix A) reveal that compounds 2–4 showed high anti-TMV activities with inhibition rates of 44.4%, 66.5%, and 52.3%, respectively. These rates are higher than those of the positive control (with inhibition rate of 32.8%). Compounds 1 and 5 also showed potential anti-TMV activities with inhibition rates of 26.5% and 31.8%, respectively. In addition, the IC_50_ values of five new compounds were also obtained, and the results (Table 4) reveal that compounds 1–5 showed IC_50_ in the range of 14.2~55.3 µM. 

Since compounds 2–4 exhibit potential anti-TMV activities, the protective effects of compounds 2–4 on TMV were also evaluated by pretreating the tobacco plant with 20 μM solutions of compounds or a solution of DMSO for 6 h before inoculation with TMV. The results show that compounds 2–4 demonstrated protective effects against host plants with inhibition rates of 40.8%, 68.2%m and 49.3% (Appendix A), respectively. These results indicated that pretreatment with compounds 2–4 could greatly increase the resistance of the host plant to TMV infection.

### 2.3. Anti-Rotavirus Activities

In order to study whether the chromones and indole alkaloids from the stem bark of *C. alata* had more broad antiviral activities, compounds 1–5 were also tested for their anti-rotavirus activity. Their ability to prevent the cytopathic effects of rotavirus in MA104 cells was tested according to our previous literatures [23], and their effects were measured in parallel with the determination of antiviral activity using ribavirin as a positive control. The results (Table 5) revealed that compounds 1–5 exhibited potential anti-rotavirus activities with therapeutic index (TI) values in the range of 9.75~15.3.

## 3. Experimental Section

### 3.1. General Experimental Procedures

UV spectra were obtained using a Shimadzu UV-1900 spectrophotometer. A Bio-Rad FTS185 spectrophotometer was used for scanning IR spectra. ^1^D- and ^2^D- NMR spectroscopic data were recorded on a DRX-500 NMR spectrometer with TMS as internal standard, and chemical shifts (*δ*) are expressed in ppm with reference to the TMS signal. ESIMS and HRESIMS analyses were measured using Agilent 1290 UPLC/6540 Q-TOF mass spectrometer. Semi-preparative HPLC was performed using an Agilent 1260 preparative liquid chromatograph with Zorbax PrepHT GF (2.12 × 25 cm) or Venusil MP C_18_ (2.0 × 25 cm) columns. Column chromatography was performed using silica gel (200–300 mesh, Qing-dao Marine Chemical, Inc., Qingdao, China), Lichroprep RP-18 gel (40~63 μm, Merck, Darmstadt, Germany), Sephadex LH-20 (Sigma-Aldrich, Inc., St. Louis, MO, USA), or MCI gel (75~150 μm, Mitsubishi Chemical Corporation, Tokyo, Japan). Column fractions were monitored by TLC, visualized by spraying with 5% H_2_SO_4_ in ethanol and heating.

### 3.2. Plant Materials 

The stems of *C*. *alata* Linn. were collected from Yingjiang County, Dehong Prefecture of Yunnan Province, People’s Republic of China, in September 2020. The identification of plant material was verified by Prof. Ning Yuan. A voucher specimen (Ynni-12-20-52) was deposited in Key Laboratory of Chemistry in Ethnic Medicinal Resources, Yunnan Minzu University, P. R. China.

### 3.3. Extraction and Isolation

The air-dried and powdered *C*. *alata* (4.5 kg) were extracted four times with 70% aq. Me_2_CO (4 × 8.0 L) and filtered. The filtrate was partitioned between EtOAc and 3% tartaric acid. The EtOAc-soluble acid part (142.2 g) were applied to silica gel (200–300 mesh) column chromatography, eluting with CHCl_3_/MeOH gradient system (10:0, 9:1, 8:2, 7:3, 6:4, 5:5) to give six fractions A-F. Further separation of fraction B (9:1, 23.5 g) by silica gel column chromatography, eluted with CHCl_3_/Me_2_CO (9:1–2:1), yielded sub-fractions B1–B7. Sub-fraction B2 (8:2, 3.58 g) was subjected to silica gel column chromatography using petroleum ether/Me_2_CO, and then semi-preparative HPLC (48% MeOH/H_2_O, flow rate 20 mL/min) to give compounds 8 (15.2 mg) and 9 (22.6 mg). Sub-fraction B3 (7:3, 6.25 g) was subjected to silica gel column chromatography using petroleum ether/Me_2_CO, and then semi-preparative HPLC (55% MeOH/H_2_O, flow rate 20 mL/min) to give compounds 4 (23.2 mg) and 7 (21.5 mg).

Then, the above aqueous layer was adjusted to pH 10 with saturated NaOH aq. and extracted with EtOAc again. The EtOAc-soluble alkaloidal materials (28.4 g) were applied to silica gel (200–300 mesh) column chromatography, and eluted with CHCl_3_/MeOH gradient system (10:0, 9:1, 8:2, 7:3, 6:4, 5:5) to give six fractions A-F. Further separation of fraction B (9:1, 6.22 g) by silica gel column chromatography, eluted with CHCl_3_/Me_2_CO (9:1–2:1), yielded sub-fractions B1–B7. Sub-fraction B1 (9:1, 1.15 g) was subjected to silica gel column chromatography using petroleum ether/Me_2_CO, and then semi-preparative HPLC (78% MeOH/H_2_O, flow rate 20 mL/min) to give compound 5 (16.4 mg). Sub-fraction B2 (8:2, 2.65 g) was subjected to silica gel column chromatography using petroleum ether/Me_2_CO, and then semi-preparative HPLC (55% MeOH/H_2_O, flow rate 20 mL/min) to give compound 6 (16.4 mg). Sub-fraction B3 (7:3, 1.92 g) was subjected to silica gel column chromatography using petroleum ether/Me_2_CO, and then semi-preparative HPLC (48% MeOH/H_2_O, flow rate 20 mL/min) to give compounds 1 (19.4 mg) and 3 (21.5 mg). Sub-fraction B4 (6:4, 2.65 g) was subjected to silica gel column chromatography using petroleum ether/Me_2_CO, and then semi-preparative HPLC (48% MeOH/H_2_O, flow rate 20 mL/min) to give 2 (21.5 mg).

### 3.4. Spectroscopic Data

*4-Methoxy-2,6-dimethylisoindolin-1-one* (1): C_11_H_13_NO_2_, obtained as yellow oil; UV (MeOH) *λ*_max_ nm (log *ε*) 210 (4.06), 255 (3.58), and 292 (3.12); IR (KBr) *ν*_max_ 3084, 2946, 1658, 1614, 1568, 1459, 1358, 1136, 1060, 860 cm^−1^; ^1^H NMR and ^13^C NMR data (CDCl_3_, 500 and 125 MHz), see Table 1; positive ESIMS *m/z* 214 [M+Na]^+^, positive HRESIMS *m/z* 214.0851 [M + Na]^+^ (calcd. 214.0844 for C_11_H_13_NNaO_2_).

*6-(Hydroxymethyl)-4-methoxy-2-methylisoindolin-1-one* (2): C_11_H_13_NO_3_, Obtained as yellow gum; UV (MeOH) *λ*_max_ nm (log *ε*) 210 (4.02), 258 (3.62), and 295 (3.15); IR (KBr) *ν*_max_ 3412, 3068, 2938, 1656, 1615, 1564, 1460, 1350, 1156, 1064, 902 cm^−1^; ^1^H NMR and ^13^C NMR data (CDCl_3_, 500 and 125 MHz), see Table 1; positive ESIMS *m/z* 230 [M+Na]^+^, positive HRESIMS *m/z* 230.0790 [M+Na]^+^ (calcd. 230.0793 for C_11_H_13_NNaO_3_).

*6-(3-Hydroxypropyl)-4-methoxy-2-methylisoindolin-1-one* (3): C_13_H_17_NNaO_3_, obtained as yellow gum; UV (MeOH) *λ*_max_ nm (log *ε*) 210 (4.10), 260 (3.47), and 292 (3.03); IR (KBr) *ν*_max_ 3426, 3059, 2942, 1659, 1612, 1562, 1439, 1358, 1168, 1060, 895 cm^−1^; ^1^H NMR and ^13^C NMR data (CDCl_3_, 500 and 125 MHz), see Table 1; positive ESIMS *m/z* 258 [M+Na]^+^, positive HRESIMS *m/z* 258.1113 [M+Na]^+^ (calcd. 258.1106 for C_13_H_17_NNaO_3_).

*7-(Hydroxymethyl)-2,2-dimethyl-8-(2-oxopropyl)chroman-4-one* (4): C_15_H_18_O_4_, obtained as yellow gum; UV (MeOH) max (log *ε*) 212 (4.26), 257 (3.86), 354 (3.18) nm; IR (KBr) *ν*_max_ 3409, 2932, 2869, 1718, 1672, 1618, 1550, 1439, 1372, 1142, 874 cm^−1^; ^1^H NMR and ^13^C NMR data (CDCl_3_, 500 and 125 MHz), see Table 2; negative ESIMS *m/z* 261 [M-H]^-^; negative HRESIMS *m/z* 261.1120 [M-H]^-^ (calcd. 261.1127 for C_15_H_17_O_4_).

*2-((3,4-Dihydro-2,2-dimethyl-4-oxo-8-(2-oxopropyl)-2H-chromen-7-yl)methyl)-4-methoxy-6-methyl-isoindolin-1-one* (5): C_25_H_27_NO_5_, obtained as yellow gum; UV (MeOH) max (log *ε*) 216 (4.32), 268 (3.86), 360 (3.42) nm; IR (KBr) *ν*_max_ 3064, 2957, 2846, 1726, 1675, 1658, 1614, 1574, 1538, 1462, 1357, 1243, 1169, 1050, 963, 852 cm^−1^; ^1^H NMR and ^13^C NMR data (CDCl_3_, 500 and 125 MHz), see Table 3; positive ESIMS *m/z* 444 [M+Na]^+^; positive HRESIMS *m/z* 444.1782 [M+Na]^+^ (calcd. 444.1787 for C_25_H_27_NNaO_5_).

### 3.5. Anti-TMV Assays

The anti-TMV activities were tested using the half-leaf method according to our previous studies [21,22], and Ningnanmycin (C_16_H_23_O_8_N_7_, CAS No. 156410-09-2), a commercial cytosine nucleoside peptide type antibiotics for plant disease in China, was used as a positive control. The virus was inhibited by mixing with the solution of tested compounds (20 μM in DMSO). After 30 min, the mixture was inoculated on the left side of the leaves of *N. glutinosa*, whereas the right side of the leaves was inoculated with the mixture of DMSO solution with the virus as control. The local lesion numbers were recorded 3–4 days after inoculation. Three repetitions were conducted for each compound. The inhibition rates were calculated according to the formula: inhibition rate (%) = [(C − T)/C] × 100%(1)
where C is the average number of local lesions of the control and T is the average number of local lesions of the treatment. Ningnanmycin (20 μM in DMSO), a commercial virucide for plant disease in China, was used as a positive control.

For compounds with significant activities in half-leaf method assays, their protective effects on TMV were also evaluated by pretreating a tobacco plant with 20 μM solutions of compounds in DMSO, or a solution of DMSO for 6 h before inoculation with TMV.

### 3.6. Anti Rotavirus Assay

The anti rotavirus activities were tested according to our previous literature [23]. Human rotavirus Wa group was used to infect the cell culture MA104 in vitro, the 50% cytotoxicity concentration (CC_50_) and half maximal effective concentration (EC_50_) were evaluated, and ribavirin (C_8_H_12_N_4_O_5_, CAS No. 1646818-35-0, a broad-spectrum antiviral drug) was used as a positive control. MA-104 cells (1 × 10^5^ cells per well) were grown in 96-well plates for 48 h. The media were removed and replaced by new media containing serial dilutions of compounds under test. After incubation for 72 h, the media were discarded, and 5.0 μL of MTT solution was added to each well. Plates were then incubated at 37 °C for 4 h. The solution was removed, and 100 μL of 0.04 mol/L HCl-isopropanol was added to each well to dissolve formazan crystals. Using a microplate reader, the absorbance of each well was measured at 540 nm. After subtracting the background absorbance at 655 nm, the 50% CC_50_ of each compound was estimated by regression analysis.

In the mixed treatment assay, each compound was mixed with a 0.01 multiplicity of infection (MOI) of the rotaviruses at various concentrations (1–160 μg/mL) and incubated at 4 °C for 1 h. The mixtures were inoculated in triplicates onto near confluent MA-104 cell monolayers (1 × 10^5^ cells per well) for 1 h with occasional rocking. The solution was removed, and the cells were replaced with eagles minimum essential medium (EMEM) containing 1 μg/mL trypsin. The cells were incubated for 72 h at 37 °C under 5% CO_2_ atmosphere until the cells in the control showed complete viral cytopathic effect (CPE) by light microscopy. EC_50_ was estimated by regression analysis.

## 4. Conclusions

With the aim of continuing to explore bioactive metabolites from Cassia Linn, chemical investigations on *C. alata* were carried out. As expected, three new indole alkaloids, alataindoleins A–C (1–3); one new chromone, alatachromone A (4); and a new dimeric chromone-indole alkaloid, alataindolein D (5) were isolated. To the best of our knowledge, alataindolein D (5) represents a new type of dimeric alkaloid with an unusual N-2−C-16’ linkage, which is biogenetically derived from a chromone and an indole alkaloid via an intermolecular nucleophilic substitution reaction. The anti-TMV assay revealed that compounds 2–4 showed high anti-TMV activities with inhibition rates of 44.4%, 66.5%, and 52.3%, respectively. These rates are higher than that of positive control. In addition, compounds 1–5 exhibited potential anti-rotavirus activities with therapeutic index (TI) values in the range of 9.75~15.3. The successful isolation and structure identification of the above new compounds provide materials for the screening of antivirus drugs and contribute to the development and utilization of *C. alata.*

## Figures and Tables

**Figure 1 molecules-27-03129-f001:**
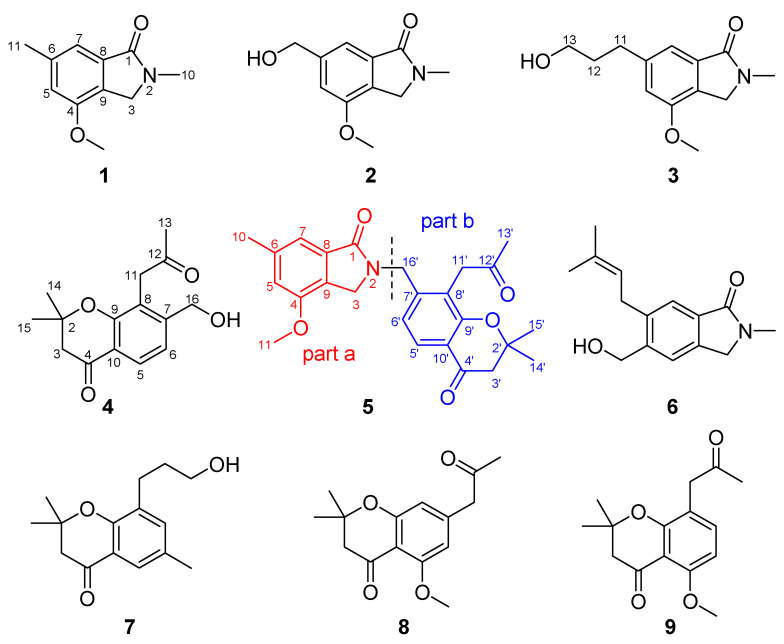
The structure of compounds **1**–**9** from *C. alata*.

**Figure 2 molecules-27-03129-f002:**
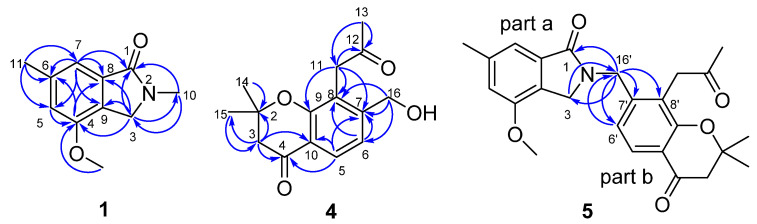
The key HMBC correlations of compounds **1**, **4**, and **5**.

**Figure 3 molecules-27-03129-f003:**
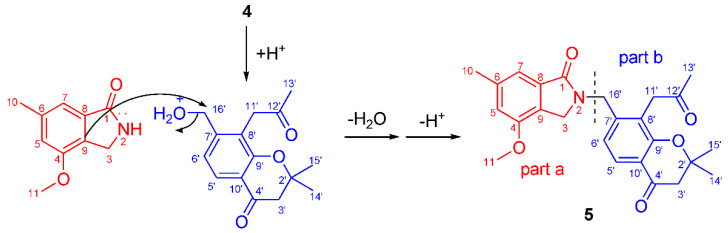
Plausible biosynthetic pathway of compound **5**.

**Table 1 molecules-27-03129-t001:** ^1^H and ^13^C NMR data for compounds **1**–**3** (in CDCl3, 500 and 125 MHz).

No.	1		2		3	
	*δ* _C_	*δ*_H_ (m, *J*, Hz)	*δ* _C_	*δ*_H_ (m, *J*, Hz)	*δ* _C_	*δ*_H_ (m, *J*, Hz)
1	168.2 s		168.5 s		168.6 s	
3	43.3 t	4.29 s	43.6 t	4.31 s	43.8 t	4.30 s
4	158.6 s		158.8 s		158.3 s	
5	115.3 d	6.77 (d) 1.6	114.0 d	6.85 (d) 1.8	114.9 d	6.82 (d) 1.8
6	138.2 s		140.7 s		137.6 s	
7	120.2 d	7.13 (d) 1.6	118.3 d	7.35 (d) 1.8	120.8 d	7.18 (d) 1.8
8	132.9 s		133.4 s		131.7 s	
9	126.6 s		128.7 s		126.1 s	
10	33.8 q	2.86 s	34.1 q	2.84 s	33.7 d	2.87 s
11	23.7 q	2.37 s	66.2 t	4.67 s	30.8 t	2.66 (t) 7.8
12					36.7 t	1.85 m
13					63.7 t	3.59 (t) 6.6
-OMe	56.2 q	3.78 s	56.2 q	3.78 s	56.1 q	3.79 s

**Table 2 molecules-27-03129-t002:** ^1^H NMR and ^13^C NMR data for compound **4** (in CDCl_3_, 500 and 125 MHz).

No.	*δ*_C_ (m)	*δ*_H_ (m, *J*, Hz)	No.	*δ*_C_ (m)	*δ*_H_ (m, *J*, Hz)
2	78.1 s		9	157.8 s	
3	50.3 t	2.64 s	10	118.7 s	
4	191.6 s		11	45.5 t	3.94 s
5	128.7 d	7.56 (d) 8.2	12	205.3 s	
6	117.0 d	6.82 (d) 8.2	13	29.9 q	2.27 s
7	143.8 s		14,15	26.4 q	1.57 s
8	127.2 s		16	63.7 t	4.46 s

**Table 3 molecules-27-03129-t003:** ^1^H and ^13^C NMR data for compound **5** (CDCl_3_, 500 and 125 MHz).

No.	*δ* _C_	*δ*_H_ (m, *J*, Hz)	No.	*δ* _C_	*δ*_H_ (m, *J*, Hz)
1	168.9 s		5	127.8 d	7.49 (d) 8.2
3	43.1 t	4.25 s	6	118.5 d	6.63 (d) 8.2
4	158.3 s		7	139.6 s	
5	115.1 d	6.73 (d) 1.6	8	129.2 s	
6	138.4 s		9	156.4 s	
7	120.4 d	7.08 (d) 1.6	10	117.6 s	
8	132.3 s		11	45.0 t	3.95 s
9	126.2 s		12	205.8 s	
10	23.4 q	2.42 s	13	30.3 q	2.25 s
2	78.4 s		14,15	26.2 q	1.61 s
3	47.9 t	2.66 s	16	41.7 t	4.33 s
4	192.2 s		-OMe	56.2 q	3.78 s

**Table 4 molecules-27-03129-t004:** Anti-TMV activity of compounds **1**–**5** on *Nicotiana glutinosa* leaf *^a^*.

No.	% Inhibition at 20 µM	IC_50_ (µM)
1	26.5 ± 2.8	55.3
2	44.4 ± 3.5	22.6
3	66.5 ± 4.0	14.2
4	52.3 ± 3.6	17.9
5	31.8 ± 3.3	43.4
Ningnanmycin	32.8 ± 3.0	37.2

*^a^* All results are expressed as mean ± SD; *n* = 3.

**Table 5 molecules-27-03129-t005:** Anti-rotavirus activity of compounds **1**–**5**.

No.	CC_50_ (µg/mL)	EC_50_ (µg/mL)	TI (CC_50_/EC_50_)
1	142.4	14.6	9.75
2	175.2	14.8	11.8
3	208.3	13.6	15.3
4	192.5	15.5	12.4
5	153.6	12.8	12.0
Ribavirin	234.5	12.2	19.2

CC_50_: mean (50%) value of cytotoxic concentration; EC_50_: mean (50%) value of effective concentration; TI: therapeutic index, CC_50_/EC_50_.

## Data Availability

The data presented in this study are available on request from thecorresponding author.

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
