# Peer review of "Indole Alkaloids and Chromones from the Stem Bark of Cassia alata and Their Antiviral Activities"

_molecules, 2022, doi:10.3390/molecules27103129_

Round 1

Reviewer 1 Report

I believe that the article is of interest because it deals with the isolation of five new and bioactive compounds from the stem bark of C. alata. It is important to highlight that the assignment of the structures was adequate making use of the conventional spectroscopy and spectrometry techniques for these cases.

Some minor corrections and suggestions to the authors to help improve the document are highlighted in the PDF attached to this evaluation. Although they are minor corrections, the authors must take them into account to improve the understanding of some parts of the text of the document.

Author Response

Dear Prof. Shawn Huang and the Reviewers,

Thank you for having evaluated our manuscript for its suitability for publication in Molecules. We have read the comments carefully from reviewers. These comments can help us to improve the quality of the manuscript. We had carefully revised the manuscript according to the referees’ suggestions and your requirements. All of the revisions made in our manuscript were highlighted in yellow color for your convenience to check. Thanks for your careful work and we look forward to hearing from your further information about its disposition.

Sincerely,

Prof. Zhou

E-mail address: jszxtg_2015@163.com

For reviewer’s comments 1

  1. The corrections and suggestion that reviewer pointed out and other we found, had been carefully revised in the revised manuscript.

Reviewer 2 Report

In this manuscript, the authors successfully isolated and identified three new indole alkaloids, alataindoleins A–C (1–3), one new chromone, alatachromone A (4), and a new dimeric chromone-indole alkaloid, alataindolein D (5) from the extract of stem bark of Cassia alata plant. Anti-tobacco mosaic virus (TMV) and anti-rotavirus activities were measured for all of these new products (1~5), some of which exhibited potential inhibition compared to the corresponding positive controls.

The manuscript is well written in general. However, the following concerns should be addressed before the acceptance for publishing in Molecules journal.

1. For the protective effects of compounds 2–4 on TMV by pretreating the tobacco plant, is it possible to increase a positive control, such as ningnanmycin, for comparsion?

2. What is the proton peak for chemical shift at 4.9 ppm in 1H NMR supplementary?

3. Please double check and correct the typos through the manuscript, such as in line 29 and 330, "new" instead of "now"; line 183 and 201, "values" instead of "valves"; in section 3.3, choose one name between "acetone" and "Me2CO", and so on.

Author Response

Dear Prof. Shawn Huang and the Reviewers,

Thank you for having evaluated our manuscript for its suitability for publication in Molecules. We have read the comments carefully from reviewers. These comments can help us to improve the quality of the manuscript. We had carefully revised the manuscript according to the referees’ suggestions and your requirements. All of the revisions made in our manuscript were highlighted in yellow color for your convenience to check. Thanks for your careful work and we look forward to hearing from your further information about its disposition.

Sincerely,

Prof. Zhou

E-mail address: jszxtg_2015@163.com

For reviewer’s comments 2

  1. For the protective effects of compounds 2-4 on TMV by pretreating the tobacco plant, The Figure (Figure S18, e) for pretreating the tobacco plant with ningnanmycin (positive control) had been added in the revised Supplementary Materials.
  2. the proton peak for chemical shift at 4.92-4.96 ppm is the water or active hydrogen peaks.
  3. We had carefully checked and amended the spelling mistakes that reviewer pointed out and other we found, which marked up by yellow highlights in revised manuscript.

If this manuscript needed some further revisions, please contact us as soon as possible without any hesitate

Round 2

Reviewer 2 Report

Please double check the Figure 3, the attack arrow should be stated from lone pair electrons of nitrogen.

Overall, the authors have done all the necessary correction and now the manuscript could be accepted in its current version.